# A Novel Image-Classification-Based Decoding Strategy for Downlink Sparse Code Multiple Access Systems

**DOI:** 10.3390/e25111514

**Published:** 2023-11-04

**Authors:** Zikang Chen, Wenping Ge, Juan Chen, Jiguang He, Hongliang He

**Affiliations:** 1College of Computer Science and Technology, Xinjiang University, Urumqi 830046, China; 2Signal Detection and Processing Key Laboratory, Urumqi 830046, China; 3Technology Innovation Institute, Abu Dhabi P.O. Box 9639, United Arab Emirates; 4Centre for Wireless Communications, University of Oulu, 90014 Oulu, Finland; 5College of Mechanical Engineering and Electronic Information, China University of Geosciences, Wuhan 430074, China

**Keywords:** sparse code multiple access (SCMA), deep learning (DL), signal detection, bit error rate (BER)

## Abstract

The introduction of sparse code multiple access (SCMA) is driven by the high expectations for future cellular systems. In traditional SCMA receivers, the message passing algorithm (MPA) is commonly employed for received-signal decoding. However, the high computational complexity of the MPA falls short in meeting the low latency requirements of modern communications. Deep learning (DL) has been proven to be applicable in the field of signal detection with low computational complexity and low bit error rate (BER). To enhance the decoding performance of SCMA systems, we present a novel approach that replaces the complex operation of separating codewords of individual sub-users from overlapping codewords using classifying images and is suitable for efficient handling by lightweight graph neural networks. The eigenvalues of training images contain crucial information, such as the amplitude and phase of received signals, as well as channel characteristics. Simulation results show that our proposed scheme has better BER performance and lower computational complexity than other previous SCMA decoding strategies.

## 1. Introduction

### 1.1. Background

In the thriving era of 5G wireless communication, the pursuit of high spectrum efficiency, massive connectivity, and low latency has intensified significantly. Orthogonal multiple access (OMA) demonstrates high performance during 3G and 4G communication, which is attributed to its straightforward and effective system architecture. However, in scenarios with massive connectivity, the requirement for the orthogonality of user carrier frequencies by OMA-based systems somewhat exacerbates the shortage in the spectrum. Therefore, it is challenging to achieve 5G’s high communication requirements by solely depending on OMA. Non-orthogonal multiple access (NOMA) [1] has, consequently, steadily evolved, which can be classified into two categories: power-domain NOMA, which achieves spectrum resource multiplexing by judiciously allocating power, and code-domain NOMA, which realizes the sharing of all wireless resources by assigning non-orthogonal extension codes to each user and identifying them with a low-complexity scheme. Among the various code-domain schemes of NOMA, sparse code multiple access (SCMA) [2], with a high overload capacity and interference resistance, stands out. SCMA allows multiple users to broadcast concurrently on the same time–frequency resources, relieving the increasing strain of spectrum limitations and enhancing the channel capacity in 5G wireless communication.

### 1.2. Related Work and Motivation

Each user in the SCMA system is assigned a unique known codebook, in which the information bits are multiplied by the mapping matrix and mapped into multi-dimensional codewords [2]. Therefore, the design of the codebook is a crucial factor that directly influences the performance of SCMA systems. The authors of [3] introduced the first codebook for SCMA, which laid the theoretical foundation for subsequent SCMA codebook designs. Improvements were made to the codebook design in [4], and a temporary codebook that can be automatically updated with reverse derivation was developed in [5]. Ref. [6] proposed a method named simplified decomposition of the superposed constellation (S-DCSC), which gained lager superposed symbols in the receiver than other codebook design schemes. The codebook design techniques are continually evolving to facilitate a more effective SCMA system.

The receiver design also plays an essential role in SCMA performance. The same time–frequency resources can be shared by multiple users in an SCMA system. As a result, it requires a more complex receiver design than an OMA-based system. The traditional decoding method for SCMA relies on the message passing algorithm (MPA) [7], which has a high computational complexity. Ref. [8] removed the exponential operations in the message updates in order to lower the computational complexity; however, this had a major negative impact on the bit error rate (BER) performance. Ref. [9] proposed a shuffled MPA (SMPA) that employs a different message updating approach, reducing the computational complexity and concurrently improving the BER performance. Based on the works of [9], a low-complexity MPA algorithm (TB-RMPA) was proposed in [10]. TB-RMPA further decreased the computational complexity, but showed a minor decline in terms of the BER performance. Although these MPA-based derivative algorithms have improved the strategy of message iteration, they still require a considerable number of cyclic iterations, which struggle to meet the low latency requirements of the decoder.

Deep learning (DL) technology is a revolutionary technique with versatile applications across various fields. Although the process of DL-based model training can be time-consuming, the achieved high efficiency and low complexity after training, along with its strong recognition and classification capabilities, make it highly suitable for signal detection systems [11].

It has been shown that the processing of complex received-signal classification in SCMA systems is compatible with powerful deep neural networks (DNNs) [12], leading to the growing popularity of DL-based SCMA decoding techniques. Compared with the approach in [13], which integrated MPA logic into a DNN, both [14,15] forwent MPA logic altogether in the decoder and instead focused on optimizing the DNN structure and selecting appropriate functions. Building upon the work of [15], Ref. [16] explored incorporating prior knowledge migration before decoding, resulting in enhanced performance. However, the aforementioned techniques may not fully utilize the potential of DNNs’ capability for feature extraction in data categorization tasks.

In this paper, our main emphasis is on enhancing the performance of SCMA decoders. For a downlink SCMA system with *J* independent users, *M* different kinds of transmission symbols (e.g., 0,1,…,M−1), and *K* orthogonal resource elements (REs), we propose a novel decoding scheme based on a lightweight graph neural network (GNN) with high efficiency and accuracy. The proposed decoding scheme converts the complex process of signal detection from received overlapping codewords that are affected by noise and fading into an easily implemented image classification procedure. The lightweight architecture of MobileNet [17] facilitates its deployment on mobile devices. Additionally, MobileNet’s design makes it simple to expand to various widths and depths, catering to varying computational resources and accuracy requirements. Therefore, a modified MobileNet model was adopted as the classification network.

### 1.3. Contributions

The main contributions of this paper are summarized as follows.
We raise the idea of sub-coordinate systems, which make it possible to map information from received signals and transmission channels into a *K*-Polygon diagram. The differences in types of *K*-Polygon diagrams can be ultimately attributed to the differences in the corresponding transmission symbol combinations, enabling the replacement of the MPA in SCMA decoding with image classification technology.We propose the concept of a dynamic dataset with automatic labeling, which can generate the enormous *K*-Polygon diagrams needed for training. This method avoids the necessity for offline dataset hand labeling, which, to some extent, reduces the training time of the model, facilitating its faster deployment in new channel environments.Compared to other SCMA decoders, our decoder has better BER performance and lower computational complexity, offering a novel perspective for achieving signal detection tasks by directly classifying the generated *K*-Polygon diagrams.

### 1.4. Organization

The remainder of this paper is organized as follows. Section 2 introduces the downlink SCMA system model. Section 3 describes our decoding scheme for tackling signal detection tasks through the image classification process in detail. Section 4 presents and evaluates the simulation results. Finally, Section 5 draws the conclusions.

## 2. SCMA System

We consider a downlink SCMA system with *J* independent users and *K* orthogonal REs (overloading factor λ = J/K>1). The quantities of orthogonal REs occupied by a user and neighbors on an orthogonal RE are $N_{r}$ and $N_{u}$, respectively. The definitions of the parameters in the downlink SCMA system mentioned in this paper are presented in Table 1.

The transmitter encodes the $\log_{2}\left(M\right)$ data bits of user *j* and maps them to a *K*-dimensional sparse codeword, $x_{j}$. The *K*-dimensional non-sparse overlaid codeword is then broadcast over *K* REs after the *K*-dimensional sparse codewords of *J* users are multiplexed.

At the transmitter, the transmitted signal r can be represented as follows:(1)$$r=\sum_{j=1}^{J}x_{j},$$
where $x_{j}={\left(x_{j1},x_{j2},\ldots ,x_{jK}\right)}^{T}$ is the codeword in the codebook $C_{j}$ with size *M* of user *j*, and $C_{j}$ is derived from the total codebook $C=\left\{C_{j},1\leq j\leq J\right\}$ constructed based on a multi-dimensional constellation.

At the receiver, the signal received by the user *j* can be expressed as follows:(2)$$y_{j}=\mathrm{diag}\left(h_{j}\right)r+n_{j},$$
where $h_{j}={\left(h_{j1},h_{j2},\ldots ,h_{jK}\right)}^{T}$ is the channel vector between the transmitter and the user *j*, and $n_{j}={\left(n_{j1},n_{j2},\ldots ,n_{jK}\right)}^{T}$ is the additive white Gaussian noise (AWGN) with zero mean and variance $\sigma^{2}$. For the downlink Rayleigh fading channel, $h_{jk}$ experienced by the transmitted signal on the *k*-th RE is the radial component of the sum of two Gaussian distributed random variables, whereas for the AWGN channel, $h_{jk}$ is constant. A basic downlink SCMA system model is presented in Figure 1, where *J* = 6, *K* = 4, and *M* = 4.

## 3. Our Proposed Method

The objective of the downlink SCMA system receivers is to precisely reconstruct the $\log_{2}\left(M\right)$ data bits of each of the *J* users from the received signal $y_{j}$ at the *K* REs, which is defined in Equation (Equation 2). The non-orthogonality between user carrier frequencies makes the design of SCMA system receivers more challenging compared with OMA-based systems. At the same time, noise interference and unstable channel characteristics in the wireless channel furtsher increase the difficulty of designing receivers with excellent performance for SCMA systems. In order to improve the downlink SCMA systems’ decoding performance, we propose a novel image-classification-based decoding strategy that allows image classification technology to take the role of the MPA in SCMA receivers to accomplish signal detection tasks.

In this section, we provide a detailed introduction to the algorithm design of the proposed *K*-Polygon diagram classification decoder (KDCD). We take into account both the effect of the AWGN and the downlink Rayleigh fading channel on signal transmission. A modified MobileNet model is adopted as the classification network. With the exception of the first layer of the convolutional layers, which is a fully convolutional layer, all remaining convolutional layers in the classification network are depthwise separable convolutional layers. Figure 2 depicts the difference between depthwise separable convolutional layers and standard convolutional layers [17]. The network parameters are learned from a large amount of training data. In order to transform the intricate process of classifying received overlapping codewords disrupted by noise and fading into an image classification task and obtain sufficient *K*-Polygon diagrams for training, we originally propose the concepts of sub-coordinate systems and dynamic datasets with automatic labeling.

### 3.1. Sub-Coordinate System

The realization of the sub-coordinate system is accomplished in the PyTorch environment, utilizing the matplotlib module [18]. First, we establish *K* virtual sub-coordinate systems in the planar rectangular coordinate system. After that, *K* points are obtained by mapping the *K* components of the received signal $y_{j}$, both real and imaginary (e.g., Reyjk and Imyjk, respectively, represent the real and imaginary parts of $y_{jk}$, which is the *k*-th component of the received signal $y_{j}$), into the *K* sub-coordinate systems, respectively, which can be expressed as
(3)SXk,SYk=Reyjk,Imyjk,
where $\left(S_{X_{k}},S_{Y_{k}}\right)$ is the point’s coordinate in the *k*-th sub-coordinate system.

Following that, by establishing a transformation function that express the spatial relationship between each sub-coordinate system and the planar rectangular coordinate system, we can compute the precise coordinates of the *K* points in the planar rectangular coordinate system, which can be expressed as
(4)$$\left(X_{k},Y_{k}\right)=\left(A\cdot S_{X_{k}}+X_{O_{k}},A\cdot S_{Y_{k}}+Y_{O_{k}}\right),$$
where $\left(X_{k},Y_{k}\right)$ represents the coordinate of the *k*-th point in the original planar rectangular coordinate system, *A* denotes the ratio of the unit distance within the *k*-th sub-coordinate system to that of the original planar rectangular coordinate system, and $\left(X_{O_{k}},Y_{O_{k}}\right)$ is the coordinate of the origin of the *k*-th sub-coordinate system in the original plane rectangular coordinate system.

Next, for a receiver with perfect channel state information (CSI), during the single communication process, the channel vector $h_{j}$ is calculated instantaneously in real-time to evaluate the characteristics of the wireless channel [16]. Floating-point numbers in the 0 to 1 range may be converted to color values using the matplotlib.colors module and colormap class [18]. The following setting is made to the floating-point number that corresponds to the color value of the *k*-th point:(5)sighjk·ImhjkRehjk,
where $\left|h_{jk}\right|$ represents the magnitude of $h_{jk}$; Rehjk and Imhjk, respectively, represent the real and imaginary parts of $h_{jk}$; and $\mathrm{sig}\left(\cdot \right)$ refers to the sigmoid activation function.

Finally, by not displaying the plane rectangular coordinate system and sequentially connecting the *K* points, with the connecting lines during this process set to black, we obtain a closed the *K*-Polygon diagram for training as shown in Figure 3. It is worth noting that, during the process of handling the output of *K*-Polygon diagram, we output only the region of the image corresponding to the x-axis range $\left[-2s:+2s\right]$ and the y-axis range $\left[-2s:+2s\right]$. This operation eliminates excessive white space around the output *K*-Polygon diagram, thereby facilitating enhanced quantization precision. The output format of the *K*-Polygon diagram is set as a grayscale image with dimensions of 56 × 56, where “56 × 56” specifies the size of the image in terms of width and height in pixels.

The aforementioned illustrates how a specific *K*-Polygon diagram is generated. There are a total of $M^{J}$ different types of *K*-Polygon diagrams, which correspond to the $M^{J}$ types of input symbol combinations, when *J* users are taken into account, each having *M* kinds of transmission symbols.

Sub-coordinate system mapping effectively avoids the mutual interference between coordinate points caused by directly mapping the received signal components to the plane rectangular coordinate system, and further distinguishes the feature-point distribution of different types of *K*-Polygon diagrams. In addition to the above, the connecting lines in the *K*-Polygon diagram provide extra exploitable features, which are beneficial for the image classification.

### 3.2. Dynamic Dataset with Automatic Labeling

To enhance the network’s fitting effect, we first generate a large number of random input symbol combinations for the SCMA system and obtain the received signal set required for generating adequate *K*-Polygon diagrams for training. It is worth noting that weak signals caused by very low values of $E_{b}/N_{0}$ (energy per bit to noise power spectral density ratio), as well as overfitting caused by excessively high values of $E_{b}/N_{0}$, may both negatively impact the BER performance of the decoder. Therefore, in every communication process, the noise n should be randomly generated within an appropriate range of $E_{b}/N_{0}$ values. To determine the optimal $E_{b}/N_{0}$ values for the received signal set, we test the following scenarios in this paper.
S1: We choose some lower values, i.e., $E_{b}/N_{0}\in \left[2:2:6\right]$ dB.S2: We choose some higher values, i.e., $E_{b}/N_{0}\in \left[18:2:20\right]$ dB.S3: We train the model with a wide range of values, i.e., $E_{b}/N_{0}\in \left[0:2:20\right]$ dB.S4: By removing the lower and higher values, we train the model using $E_{b}/N_{0}\in \left[8:2:16\right]$ dB.

In the PyTorch environment, a significant number of cycles are typically specified to ensure that the entire image training set is traversed during each iteration of training. However, in our scheme, the image batch used in each training cycle is pre-generated on-the-fly by the corresponding component in the received signal set and shares a storage tensor.

We then develop the idea of automatically tagging *K*-Polygon diagrams. The input symbol combination corresponding to the *K*-Polygon diagram should be processed as a *J*-bit *M*-ary number which equals a decimal number. The label for the *K*-Polygon diagram should have the same size as the output of the KDCD and be represented as a one-hot tensor [19], where the element indexed by this decimal number is assigned as one individually. Thus, the operation of automatically tagging *K*-Polygon diagrams can be expressed as
(6)$$\begin{array}{c} d=\sum_{j=1}^{J}m_{j}M^{J-j},\\ b\left[d\right]=1, \end{array}$$
where *d* denotes the decimal number, $m_{j}$ is the input symbol of the *j*-th user, and b denotes the one-hot label. Since the entire *K*-Polygon diagram dataset is not present at any given time and the labeling process is automatic, we refer to the dataset used for training as the dynamic dataset with automatic labeling.

Dynamic dataset with automatic labeling circumvents the need for manual labeling of offline datasets, streamlining the process of creating complete *K*-Polygon diagram datasets and ultimately reducing the time and resource expenses associated with pre-training models.

### 3.3. Model Optimization

To classify a *K*-Polygon diagram and predict the $\log_{2}\left(M\right)$ data bits for each independent user, we train the network model parameters by minimizing the following loss function:(7)$$L\left(p,b\right)=-\sum_{i=1}^{M^{J}}b_{i}\log\left(p_{i}\right),$$
where the function $L\left(\cdot \right)$ is the well-known cross-entropy loss, $p={\left[p_{1},\ldots ,p_{M^{J}}\right]}^{T}$ is the output of MobileNet’s fully connected layer, and b is the *K*-Polygon diagram’s one-hot label of the same size as p (e.g., the corresponding input symbol combination of $b={\left[1,0,\ldots ,0\right]}^{T}$ is ${\left[0,0,\ldots ,0\right]}^{T}$).

### 3.4. Model Configuration

Although the proposed decoding scheme can be applied to larger SCMA systems, it is important to note that the design of codebooks for such systems is beyond the scope of our study. Additionally, channel estimation is not the primary focus of this research. Therefore, in this paper, we consider a basic downlink SCMA system model with *J* = 6, *K* = 4, *M* = 4 and, perfect CSI, which is consistent with the settings in [15,16]. The *A* and $\left(X_{O_{k}},Y_{O_{k}}\right)$ in Equation (Equation 4) and the *s*, which governs the range of diagram output, are set as
(8)$$\begin{array}{c} A=1,\\ \left(X_{O_{k}},Y_{O_{k}}\right)=\left\{\begin{array}{c} \left(+s,+s\right),k=1\\ \left(-s,+s\right),k=2\\ \left(-s,-s\right),k=3\\ \left(+s,-s\right),k=4 \end{array},\right.\\ s=5. \end{array}$$

The colormap, designated as “gray”, is set to determine the functional relationship between the floating-point number and color value of *K* points in the *K*-Polygon diagram. The classification network utilizes a modified MobileNet model, with the comprehensive structure illustrated in Table 2.

## 4. Simulation and Evaluation

The methods being compared in this study are all based on the basic downlink SCMA system model, with *J* = 6, *K* = 4, *M* = 4, and perfect CSI. The components of the channel vector $h_{j}$ are modeled as independently and identically distributed complex Gaussian random variables with zero mean and unit variance. And each method uses the same codebook, provided by [20]. The whole received signal set has 2 million random samples. In order to minimize the loss function in Equation (Equation 7), we adopt an adaptive motion estimation (ADAM) optimizer [21], in which the learning rate is set as 0.002. The batch size of *K*-Polygon diagrams supplied into the classification network per cycle is set as 64.

### 4.1. Choice of $E_{b}/N_{0}$ Values

Figure 4 shows the BER performance of the KDCD over the AWGN and downlink Rayleigh fading channel after it was trained using each of the aforementioned scenarios. The simulation findings show that BER performances vary under different training scenarios. The reasons for this are as follows.
S1: The chosen $E_{b}/N_{0}$ values for training are very low, resulting in excessive noise interference in signal transmission, and thereby reducing the effectiveness of training.S2: The network model is overfitted as a result of the $E_{b}/N_{0}$ values being chosen as excessively high, which decreases the BER performance of the KDCD at low $E_{b}/N_{0}$ values.S3: It is not able to prevent the detrimental effects of very low and excessively high $E_{b}/N_{0}$ values on training, which leads to a suboptimal BER performance, even though a wide range of $E_{b}/N_{0}$ values are chosen for training.S4: The best BER performance is achieved by choosing the appropriate range of $E_{b}/N_{0}$ values that prevents overfitting in the network model and enhances training efficacy.

When compared with the other scenarios, S4 is the optimum training strategy. Therefore, the noise n for training is randomly generated with different values of $E_{b}/N_{0}$: 8, 10, 12, 14, and 16 in the rest of this work.

### 4.2. BER Comparison

Figure 5 and Figure 6 compare the BER performance of our KDCD with different SCMA decoders, including a TB-RMPA [10], and deep learning decoder (DLD) [15], over the AWGN and downlink Rayleigh fading channel, respectively. According to Figure 5, Our KDCD consistently outperforms the deep learning decoder for the AWGN channel (DLD-A) [15] and TB-RMPA across different $E_{b}/N_{0}$ values. It is noteworthy that when the value of $E_{b}/N_{0}$ exceeds 10 dB, as $E_{b}/N_{0}$ increases, the BER performance of the KDCD becomes more significant. In addition, as shown in Figure 6, our KDCD also exhibits better BER performance than the TB-RMPA and deep learning decoder for the downlink Rayleigh fading channel (DLD-R) [15]. Based on this analysis, the explanations for the superior decoding performance of our proposed KDCD in various channel environments compared with other decoders are as follows.
Compared with the conventional decoding algorithm (TB-RMPA): Our KDCD can directly utilize the effective information of the received signals and channel characteristics to accomplish the decoding operations. However, the TB-RMPA relies on a large number of iterations. During every iteration, both valid and invalid information participate in the information exchange between resource nodes and user nodes, which reduces the effectiveness of message updates. In addition, our KDKD provides extra exploitable features (the features of connecting lines in the *K*-Polygon diagram), which facilitate the image classification.In contrast to previous DL-based decoders (DLDs): Our KDCD allows the neural network to leverage more features. Moreover, our KDCD can directly accomplish the signal detection tasks with image classification, fully utilizing the potential of neural networks’ capability for feature extraction.

It should be noted that MobileNet performs much better than it did in the ImageNet competition [22]. This improvement is due to the appropriate selection of $E_{b}/N_{0}$ values for training, which prevents the received signal y from being significantly affected by AWGN, leading to a high degree of similarity in the distribution of feature points on quadrilateral diagrams of the same type, which is beneficial for the image classification.

However, our KDCD shares the same communication limitations as those in [15,16]. Since channel estimation is not our main focus, our KDCD requires the strong assumption of perfect CSI. Additionally, our KDCD only contributes to the decoding of downlink SCMA systems, lacking the ability for integrated encoding and decoding, and thus requiring a well-configured codebook at the transmitter.

### 4.3. Complexity Analysis

The computational complexity of the generation of the *K*-Polygon diagram can be considered negligible compared with the computational complexity of the convolutional computation. In the *c*-th depthwise separable convolution layer, $K_{c}^{2}$, $I_{c}$, $O_{c}$, and $D_{c}^{2}$ represent the depthwise convolution kernel size, input channel number, output channel number, and output image size, respectively. In the fully convolutional layer, $K_{C}^{2}$, $I_{C}$, $O_{C}$, and $D_{C}^{2}$, respectively, denote the convolution kernel size, input channel number, output channel number, and output image size. $N_{F}$ and $N_{O}$ represent the number of neurons in the fully connected layer and the output layer, respectively. Taking into account the impact of stride, the number of Multiply–Accumulate (MAC) operations of our KDCD is as follows:(9)$$\mathrm{MAC}=\sum_{c=1}^{4}\left(K_{c}^{2}I_{c}D_{c}^{2}+I_{c}O_{c}D_{c}^{2}\right)+K_{C}^{2}I_{C}O_{C}D_{C}^{2}+N_{F}N_{O}.$$

The comparison between the KDCD, learning-aided SCMA (D-SCMA) [12], and DLD-R [15] on computational complexity is shown in Table 3. Compared with D-SCMA and DLD-R, our KDCD achieves reductions of 29.4% and 24.9%, respectively, in terms of the number of MAC operations. In addition, to demonstrate the effective reduction in communication latency achieved by our KDCD, Figure 7 compares the decoding time of different decoders in a more intuitive manner. As shown in Figure 7, our KDCD possesses the lowest communication latency, aligning with the low latency requirement of 5G wireless communication. This can be attributed to the fact that, compared with other decoders, our KDCD does not require iterations for message updates. Furthermore, our KDCD employs the modified MobileNet model with a lightweight structure as the classification network, which is highly efficient and precise.

## 5. Conclusions

We have successfully offered a novel approach for achieving signal detection tasks in SCMA receivers by transforming the intricate processes of classifying overlapping codewords disrupted by noise and fading into the image classification processes. We have raised the concepts of sub-coordinate systems and dynamic datasets with automatic labeling, which enable us to construct enough quadrilateral diagrams for network training. Our KDCD has demonstrated superior performance in terms of BER performance and computational complexity compared with other previous SCMA decoders. We will conduct further research to strive to eliminate the communication limitations of our KDCD.

## Figures and Tables

**Figure 1 entropy-25-01514-f001:**
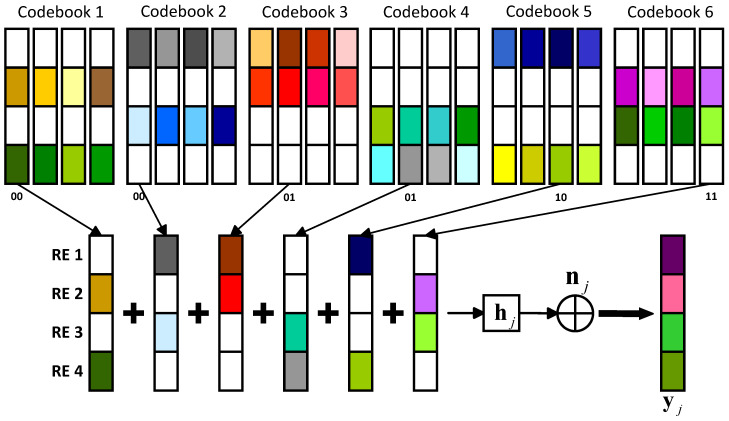
A basic downlink SCMA system model.

**Figure 2 entropy-25-01514-f002:**
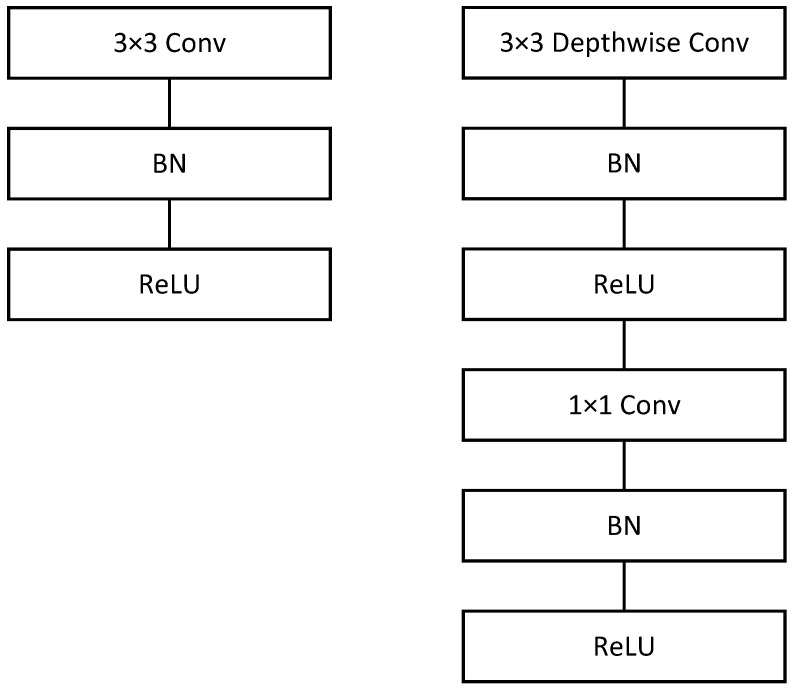
**Left**: Standard convolutional layer with BatchNorm and ReLU. **Right**: Depthwise separable convolutions with depthwise and pointwise layers followed by NatchNorm and ReLU.

**Figure 3 entropy-25-01514-f003:**
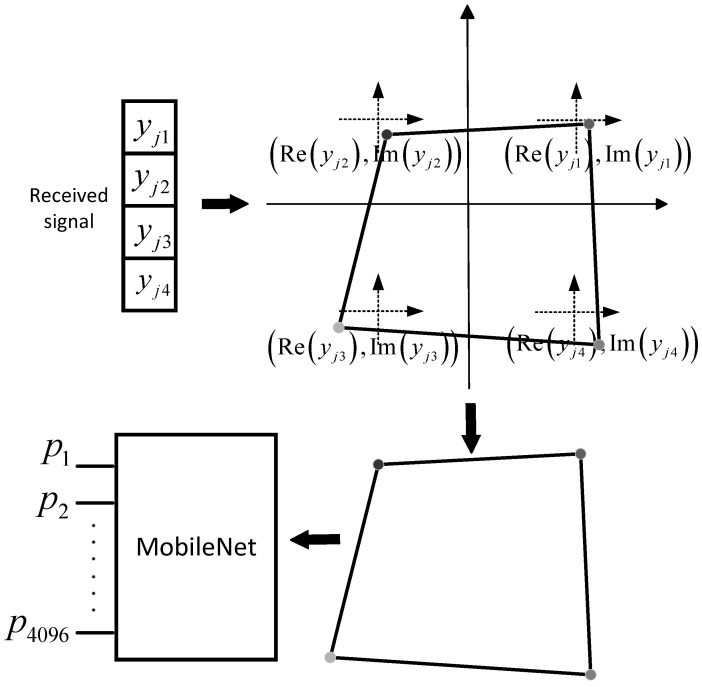
Architecture of our proposed KDCD when *J* = 6 and *K* = 4.

**Figure 4 entropy-25-01514-f004:**
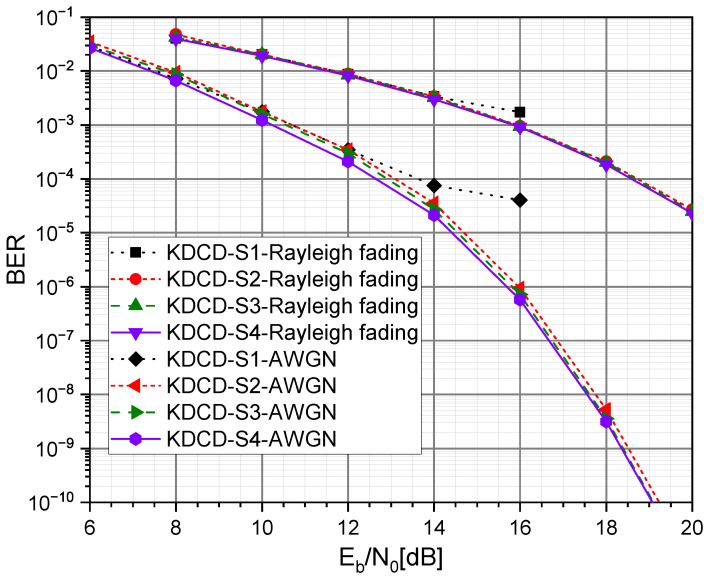
The impact of the choice of $E_{b}/N_{0}$ values for training on the BER over AWGN and downlink Rayleigh fading channel in our decoder.

**Figure 5 entropy-25-01514-f005:**
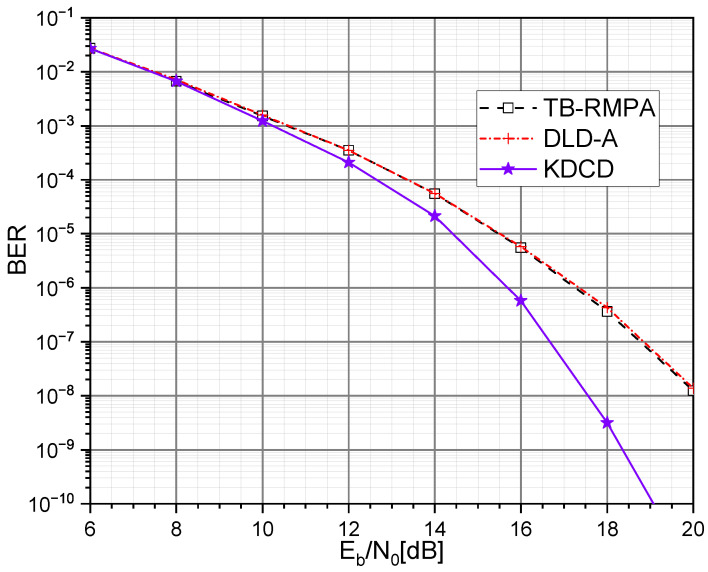
BER comparison of TB-RMPA, DLD-A, and KDCD over AWGN channel.

**Figure 6 entropy-25-01514-f006:**
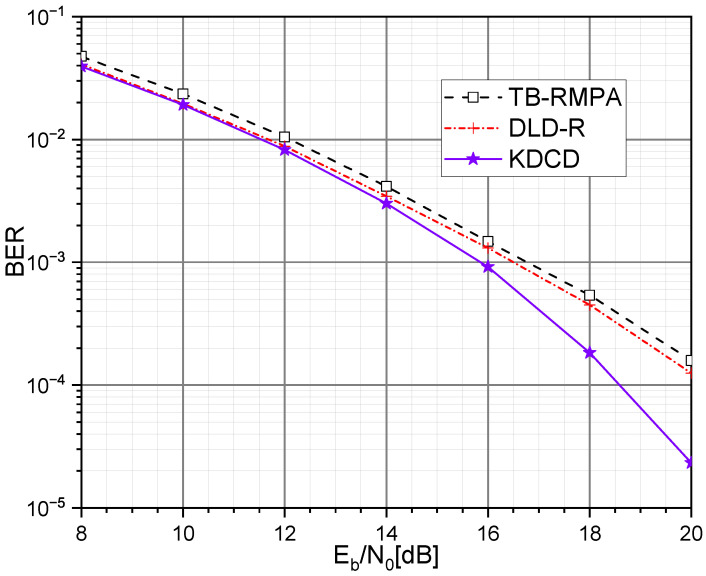
BER comparison of TB-RMPA, DLD-R, and KDCD over downlink Rayleigh fading channel.

**Figure 7 entropy-25-01514-f007:**
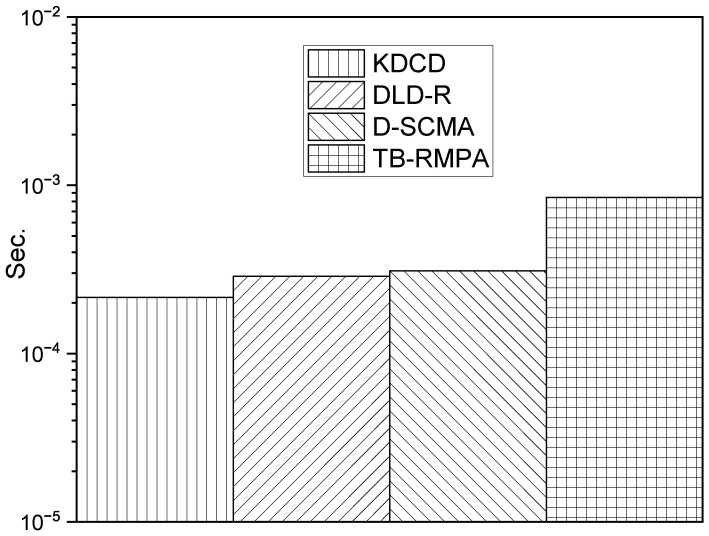
Computation time of SCMA decoders.

**Table 1 entropy-25-01514-t001:** The parameters in the downlink SCMA system.

Parameter	Definition
*J*	Number of independent users
*K*	Number of orthogonal resource elements
*M*	Size of codebook
λ	Overloading factor
$N_{r}$	Number of orthogonal resource elements occupied by a user
$N_{u}$	Number of neighbors on an orthogonal resource element

**Table 2 entropy-25-01514-t002:** Structure of modified version of the MobileNet model.

Type/Stride	Filter Shape	Input Size
Conv/s2	3 × 3 × 1 × 8	56 × 56 × 1
Conv1 dw/s1	3 × 3 × 8 dw	28 × 28 × 8
Conv1/s1	1 × 1 × 8 × 8	28 × 28 × 8
Conv2 dw/s2	3 × 3 × 8 dw	28 × 28 × 8
Conv2/s1	1 × 1 × 8 × 16	14 × 14 × 8
Conv3 dw/s1	3 × 3 × 16 dw	14 × 14 × 16
Conv3/s1	1 × 1 × 16 × 16	14 × 14 × 16
Conv4 dw/s2	3 × 3 × 16 dw	14 × 14 × 16
Conv4/s1	1 × 1 × 16 × 56	7 × 7 × 16
Avg Pool/s1	Pool 7 × 7	7 × 7 × 56
FC/s1	56 × 4096	1 × 1 × 56
Softmax/s1	Classifier	1 × 1 × 4096

**Table 3 entropy-25-01514-t003:** MAC operations of SCMA decoders.

	KDCD	DLD-R	D-SCMA
MAC	561,008	747,008	794,694

## Data Availability

Not applicable.

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
