# Peer review of "A Novel Image-Classification-Based Decoding Strategy for Downlink Sparse Code Multiple Access Systems"

_entropy, 2023, doi:10.3390/e25111514_

Round 1

Reviewer 1 Report

Comments and Suggestions for Authors

This paper designs a deep-learning decoder for the downlink of sparse code multiple access. The idea is to represent the received signal as a polygon diagram and use image classification to map the polygon pattern to transmitted symbols. Numerical results show promising performance.

Some comments and suggestions to improve the paper are as follows.

1. Calling the proposed scheme a multi-user detector can be misleading since it is the decoder for each user in the downlink, rather than a scheme for the base station in the uplink to detect multiple users.

2. The scheme relies on perfect channel state information at the users. This is a strong assumptions since channel state information is not granted for free, and channel estimation typically induces a cost that needs to be taken into account. The assumption of perfect channel state information should be clearly stated in the system model.

3. While Section 3 gives a comprehensive explanation of the scheme from a learning perspective, the communication aspect is not adequately discussed. It is suggested that the authors define clearly the communication problem first, including the aim of the receiver and the challenge of this setting. Then they present their approach to map this communication problem into a image classification problem and the tools that they use to solve the image classification.

4. This method seems to be limitted to small systems with few users, few resource elements, and small codebooks. In the experiment, the authors consider 6 users, 4 resource elements, and 4 transmission symbols. Would this be a typical communication scenario? How to scale up the solution for larger systems?

5. No analytical analysis on the performance and communication limits achievable with this approach is provided.

Comments on the Quality of English Language

Please double check for typos and improve the quality of the writing of the paper.

Author Response

Please see the attachment, thanks.

Reviewer 2 Report

Comments and Suggestions for Authors

This paper presents an SCMA decoding strategy for multi-user detection based on image classification. The reviewer believes that the introduction and motivations are clear, but the evaluation and assessment do not appear to be satisfactory.

1. In particular, the manuscript lacks discussion in Section 4. For instance, there is insufficient explanation for Figure 4, making it difficult to understand why "M" is considered the optimal training strategy. Additionally, the use of the notation "M" is confusing, given that the authors have employed italic "M" as a simulation parameter. It would be highly beneficial to include more simulation results and their corresponding explanations to emphasize the superiority of the proposed method.

2. The paper does not provide a clear explanation for why the proposed method can outperform conventional methods. As a Deep Learning-based method may involve heuristics, the authors should offer an intuitive discussion explaining why the proposed scheme outperforms conventional methods.

3. To enhance the understanding of readers, it would be helpful if the authors could provide more detailed information regarding the preliminaries of the SCMA system in Section II.

Author Response

Please see the attachment, thanks.

Round 2

Reviewer 1 Report

Comments and Suggestions for Authors

My previous comments and suggestions have been addressed. I have no further comments.

Reviewer 2 Report

Comments and Suggestions for Authors

The authors have clearly addressed all of my concerns, and I do not possess any additional comments.